# Influence of Immobilization Strategies on the Antibacterial Properties of Antimicrobial Peptide-Chitosan Coatings

**DOI:** 10.3390/pharmaceutics15051510

**Published:** 2023-05-16

**Authors:** Mariana Barbosa, Pedro M. Alves, Fabíola Costa, Cláudia Monteiro, Paula Parreira, Cátia Teixeira, Paula Gomes, Maria Cristina L. Martins

**Affiliations:** 1LAQV-REQUIMTE, Departamento de Química e Bioquímica, Faculdade de Ciências, Universidade do Porto, Rua do Campo Alegre, s/n, 4169-007 Porto, Portugalpmalves@i3s.up.pt (P.M.A.);; 2i3S—Instituto de Investigação e Inovação em Saúde, Universidade do Porto, Rua Alfredo Allen, 208, 4200-135 Porto, Portugalclaudia.monteiro@ineb.up.pt (C.M.); parreira@i3s.up.pt (P.P.); 3INEB—Instituto de Engenharia Biomédica, Universidade do Porto, Rua Alfredo Allen, 208, 4200-135 Porto, Portugal; 4Faculdade de Engenharia, Universidade do Porto, Rua Dr. Roberto Frias, s/n, 4200-391 Porto, Portugal; 5Instituto de Ciências Biomédicas Abel Salazar, Universidade do Porto, Rua de Jorge Viterbo Ferreira, 228, 4050-013 Porto, Portugal

**Keywords:** antimicrobial peptides, bacterial adhesion, biomaterials, chitosan, surface characterization, surface modification

## Abstract

It is key to fight bacterial adhesion to prevent biofilm establishment on biomaterials. Surface immobilization of antimicrobial peptides (AMP) is a promising strategy to avoid bacterial colonization. This work aimed to investigate whether the direct surface immobilization of Dhvar5, an AMP with head-to-tail amphipathicity, would improve the antimicrobial activity of chitosan ultrathin coatings. The peptide was grafted by copper-catalyzed azide-alkyne cycloaddition (CuAAC) chemistry by either its *C-* or *N-* terminus to assess the influence of peptide orientation on surface properties and antimicrobial activity. These features were compared with those of coatings fabricated using previously described Dhvar5-chitosan conjugates (immobilized in bulk). The peptide was chemoselectively immobilized onto the coating by both termini. Moreover, the covalent immobilization of Dhvar5 by either terminus enhanced the antimicrobial effect of the chitosan coating by decreasing colonization by both Gram-positive (*Staphylococcus aureus*, *Staphylococcus epidermidis*) and Gram-negative (*Escherichia coli*, *Pseudomonas aeruginosa*) bacteria. Relevantly, the antimicrobial performance of the surface on Gram-positive bacteria depended on how Dhvar5-chitosan coatings were produced. An antiadhesive effect was observed when the peptide was grafted onto prefabricated chitosan coatings (film), and a bactericidal effect was exhibited when coatings were prepared from Dhvar5-chitosan conjugates (bulk). This antiadhesive effect was not due to changes in surface wettability or protein adsorption but rather depended on variations in peptide concentration, exposure, and surface roughness. Results reported in this study show that the antibacterial potency and effect of immobilized AMP vary greatly with the immobilization procedure. Overall, independently of the fabrication protocol and mechanism of action, Dhvar5-chitosan coatings are a promising strategy for the development of antimicrobial medical devices, either as an antiadhesive or contact-killing surface.

## 1. Introduction

Targeting the early stages of infection (i.e., bacterial adhesion and colonization) is key in the fight against biofilms on biomaterials [1,2]. Thus, it comes as no surprise that a number of strategies have been emerging to develop antimicrobial surfaces, either by preventing protein adsorption (anti-fouling surfaces) or cell adhesion (anti-adhesive surfaces) or by killing bacteria (bactericidal surfaces) [3,4]. A few examples are cationic compounds [5], quorum-sensing inhibitors [6], and antimicrobial peptide-based approaches. Cationic compounds, such as polyamines or quaternary ammonium compounds [7], can interact with bacterial membranes, leading to membrane destabilization. Meanwhile, quorum sensing inhibition is based on hindering the bacterial communication system, although much is still unknown regarding the involvement of these communication pathways in cellular behavior. More recently, photodynamic therapy based on the production of reactive oxygen species by photodynamic agents, which are toxic to bacteria, and photothermal, which kills bacteria by heat, have been explored [8,9]. Although these strategies have shown some potential in vitro, they often struggle when fighting different bacterial strains [10]. Additionally, surface properties, such as surface charge and topography, also have to be considered when avoiding bacterial adhesion and subsequent growth. Unfortunately, some of the reported antimicrobial approaches have a number of downsides, such as cytotoxicity, low efficiency, and ineffective long-term stability profiles [4,11,12]. Alternatively, AMP-based strategies, which can also be used in combination with other therapeutics [13], have shown the versatility to kill multiple bacteria strains [14]. Moreover, the covalent immobilization of AMP on the surface of biomaterials has emerged as a promising strategy to overcome the aforementioned problems [3,15]. Apart from avoiding the inherent limitations of soluble AMP, namely proteolytic degradation, self-aggregation, and binding to plasma proteins [3,16], their covalent immobilization on biomaterial surfaces has the further advantage of preventing the formation of peptide concentration gradients associated with peptide-releasing therapies, thus minimizing their cytotoxicity and improving long-term stability [1]. Another advantage of AMP covalent immobilization is the prevention of biofilm formation by compromising microorganism viability after contact with the coated material, which holds promise for clinical applications [3]. Moreover, AMP immobilization can prolong residence time and provide antimicrobial action directly at the biomaterial application site [16,17]. Specifically, chitosan possesses antimicrobial activity, as the protonated amine groups are able to damage the bacterial wall [18]. Its activity has been shown to be higher for Gram-positive bacteria. However, the covalent conjugation of AMP to chitosan potentiates the polymer inherent antimicrobial activity [1,14,19]

One AMP that has been shown to potentiate the activity of chitosan is Dhvar5. Dhvar5 (LLLFLLKKRKKRKY) is a synthetic AMP with a hydrophobic region at its *N*-terminus and a cationic region at its *C*-terminus. The precise mechanism of action of soluble Dhvar5 is still not fully elucidated. Previous studies demonstrated that Dhvar5 induces leakage of intracellular content but without permanent pore formation [1,20,21]. Nevertheless, the mechanism of action of AMP grafted onto a surface is likely very different compared to soluble AMP and depends on several parameters—AMP density, exposure and orientation, immobilization chemistry, and biomaterial used. The mechanism of action of immobilized AMP may involve the formation of small perturbances on the surface of bacteria, leading to electrostatic dysregulation and cell death promotion. Peptide tethering via “click” chemistry has emerged as an interesting approach in this context [14,22,23,24]. According to our previous findings, the copper-catalyzed azide-alkyne cycloaddition (CuAAC) “click” reaction is an effective chemoselective approach to produce Dhvar5-chitosan conjugates with antimicrobial properties. Moreover, Dhvar5-chitosan ultrathin coatings (immobilization in bulk) displayed bactericidal activity with varying intensity depending on which region of the peptide was exposed, being higher when Dhvar5 was immobilized through its *C*-terminus (i.e., exposing its hydrophobic domain) [14,24].

In this study, new antimicrobial coatings were developed to evaluate if antimicrobial properties are influenced by the immobilization of antimicrobial peptides before or after the formation of the ultrathin coatings. Therefore, Dhvar5 was immobilized via CuAAC chemistry, by both termini, directly on chitosan ultrathin coatings (immobilization on film). The antimicrobial activity and surface properties of the produced coatings were compared to those of coatings fabricated using peptide-chitosan conjugates previously immobilized in bulk.

## 2. Materials and Methods

### 2.1. Dhvar5 Synthesis and Characterization

Peptide Dhvar5 (LLLFLLKKRKKRKY *C*- terminal amide, M_w_ = 1847 Da) and its derivatives, having a propargylglycine (Pra) as alkyne moiety at either the N- or the C- terminus, were synthesized by standard Fmoc/^t^Bu solid-phase peptide synthesis (SPPS), as previously reported [14,24]. For the Pra-modified derivatives, a 6-aminohexanoic acid (Ahx) spacer was inserted between the bioactive sequence and the terminal Pra residue. The complete sequences of the Pra-modified peptides (M_w_ = 2056 Da) are hereafter named *N*_t_-Ahx-Dhvar5 and *C*_t_-Ahx-Dhvar5. The crude peptides were purified by reverse-phase high-performance liquid chromatography (RP-HPLC) at a preparative scale (Hitachi-Merck LaPrep Sigma, VWR, Radnor, PA, USA), and their purity degrees were confirmed as higher than 95% by RP-HPLC at an analytical scale (Hitachi-Merck LaChrom Elite, Agilent Technologies, Santa Clara, CA, USA). Peptide M_w_ was confirmed by electrospray ionization/ion trap mass spectrometry (ESI/IT MS) on positive mode (LCQ-DecaXP LC-MS system, ThermoFinnigan, ThermoFischer Scientific, Waltham, MA, USA), similar to previous reports by us [14,24].

### 2.2. Dhvar5-Chitosan (Film) Coatings (Immobilization on Film)

#### 2.2.1. Chitosan Coatings on Gold Substrates

Commercial squid pen chitosan with high molecular weight (M_w_ = 363 ± 28 kDa) and a 94% degree of deacetylation (DD) was obtained from France Chitine. Prior to its handling, chitosan was purified by the re-precipitation method, as previously described [25]. Chitosan ultrathin coatings were prepared by the deposition of chitosan solutions (0.4% *w*/*v* in 0.1 M acetic acid) by spin-coating (Laurell Technologies Corporation, NorthWales, UK) at 9000 rotations per minute (rpm) during 1 min onto gold substrates (1 × 1 cm^2^) obtained from the “Instituto de Engenharia de Sistemas e Computadores–Microsistemas e Nanotecnologias”, Portugal (INESC-MN) [26] or onto quartz crystal microbalance gold sensors (QSX301; 78 mm^2^; 5 MHz; obtained from Biolin Scientific, Gothenburg, Sweden). Double-layered chitosan ultrathin films were produced by performing the spin-coating process twice. Once synthesized, chitosan films were neutralized with 0.1 M NaOH, rinsed with type 1 water (ultrapure water with a resistivity greater than 18 MΩ-cm, a conductivity inferior to 0.056 µS/cm and less than 50 ppb of total organic carbon), dried with a gentle stream of argon, and stored in sealed plastic Petri dishes saturated under argon.

#### 2.2.2. Conversion of Chitosan Amines into Azides (*N*_3_-Chitosan)

Chitosan ultrathin coatings were immersed for 24 h in a solution of 2 mM of imidazole-1-sulfonyl azide hydrochloride (ISA·HCl) and 1.5 mM of potassium carbonate in type 1 water, at room temperature and with orbital shaking and 100 rpm. ISA.HCl was a kind gift from Professor Fernando Albericio (University of Barcelona and University of KwaZulu-Natal). The modified films were then rinsed with type 1 water and immersed for 1 min on an ultrasound bath (Bandelin Sonorex Digitec Bath 35 kHz, BANDELIN electronic GmbH & Co. KG, Berlin, Germany) and rinsed again with type 1 water.

#### 2.2.3. Tethering of Alkyne-Modified Dhvar5 onto *N*_3_-Chitosan Coatings (Dhvar5-Chitosan Films)

Conjugation of alkyne-modified Dhvar5 derivatives onto *N*_3_-chitosan coatings to produce the *C*_t_-Dhvar5-Chitosan and the *N*t-Dhvar5-Chitosan films was conducted under standard CuAAC reaction conditions [24]. Briefly, *N*_3_-chitosan substrates were incubated with excess alkyne-modified peptide solutions (10 mg/mL) in 2 mM aqueous CuSO_4_·H_2_O Cu in the presence of 0.1 M sodium ascorbate for in situ generation of the Cu(I) catalyst. The tris(3-hydroxypropyltriazolylmethyl)amine (THPTA) Cu(I)-stabilizing ligand (0.01 M) and aminoguanidinium hydrochloride (0.1 M) were also added to the reaction mixture. The reaction was allowed to proceed for 24 h at 37 °C, under orbital shaking at 100 rpm. The modified films were then rinsed with type 1 water and immersed for 3 min on an ultrasound bath, again rinsed with type 1 water, immersed for another 2 min on an ultrasound bath, and rinsed once more with type 1 water. The modified films were then sequentially rinsed with 0.1 M aqueous ethylenediamenetetraacetic acid (EDTA) (pH = 6), 5% aqueous NaHCO_3_, and type 1 water. Finally, the modified films were immersed for an additional 2 min in an ultrasound bath and rinsed one last time with type 1 water. Samples were dried individually with a gentle stream of argon and stored in plastic Petri dishes saturated under argon.

### 2.3. Dhvar5-Chitosan (Bulk) Coatings (Immobilization in Bulk)

Dhvar5-chitosan (bulk) coatings were produced according to the work of Barbosa et al. [14]. Briefly, the first layer of chitosan gold coating was obtained by spin coating, as described above (Section 2.2.1). Then, 0.4% *w*/*v* solutions of the Dhvar5-chitosan conjugates were spin-coated as the second layer, as previously described by us [14]. Samples were then neutralized with 0.1 M NaOH and rinsed and dried as described above (Section 2.2.1).

### 2.4. Surface Characterization of Dhvar5-Chitosan Coatings

Dhvar5-chitosan ultrathin coatings were analyzed using several surface characterization techniques.

#### 2.4.1. Ellipsometry

Ellipsometry measurements were performed using an imaging ellipsometer, model EP3, from Nanofilm Surface Analysis. This ellipsometer was operated in a polarizer-compensator-sample-analyzer (PCSA) mode (null ellipsometry). The light source was a solid-state laser (λ = 532 nm). The refractive index (n = 0.7078) and extinction coefficient (k = 2.6564) of the gold substrate were determined by using a delta and psi spectrum with an angle variation between 66.5° and 76.5°. These measurements were made in four zones to correct any instrument misalignment. The thickness of the chitosan films was determined considering n = 1.54 and k = 0 for the chitosan film [27]. Results are presented as the average of three measurements on each of the two samples.

#### 2.4.2. Water Contact Angle Measurements (WCA)

WCA analyses were carried out using the sessile drop and the captive bubble methods. A contact angle measuring system from Data Physics, model OCA 15, equipped with a video CCD camera and operating with the SCA 20 software, was used for both approaches. Regarding the sessile drop method, the WCA measurements were performed as previously described [26]. Briefly, after the deposition of 4 µL drops of type 1 water, images were taken every 2 s over 300 s. Droplet profiles were fitted using the Young-Laplace formula to calculate the contact angle. The WCA of each substrate was calculated by extrapolating the time-dependent curve to zero, and the results are the average of two measurements on three independent samples. For the captive bubble method, samples were first tape glued to a microscope slide and placed for 30 min in a quartz chamber filled with type 1 water at room temperature to keep the samples hydrated prior to measurements. Afterward, 20 μL bubbles of air were injected beneath the wet ultrathin films using a J-shaped syringe. The bubble images were stored, and the respective WCA were calculated from the shape of the drop as described for the sessile drop method [28,29]. In this case, each film contact angles were the average of four bubble measurements made on three different samples.

#### 2.4.3. Fourier Transform Infrared Reflection-Absorption Spectroscopy (IRRAS)

Samples were analyzed on a Perkin Elmer FTIR spectrophotometer (Perkin Elmer, Waltham, MA, USA), model 2000, equipped with a VeeMax II Accessory (PIKE) and a liquid-nitrogen-cooled MCT detector. The sample chamber was purged with dry nitrogen for 2 min prior to and during the measurement of each sample to minimize water vapor adsorption. For each sample, a blank gold surface was used as a background. Incident light was p-polarized, and spectra were collected using the 80° grazing angle reflection mode. For each sample, 100 scans were collected at a 4 cm^−1^ resolution.

#### 2.4.4. X-ray Photoelectron Spectroscopy (XPS)

XPS measurements were performed on a Kratos Axis Ultra HAS spectrometer (Kratos Analytical, Manchester, UK) using aluminum (15 kV) as the radiation source at “Centro de Materiais da Universidade do Porto, Portugal” (CEMUP). The photoelectrons were analyzed at a take-off angle of 90° between the horizontal surface plane and the electron analyzer optics. Survey spectra were acquired over a range of 0–1350 eV with an analyzer pass energy of 80 eV. High-resolution *C1s*, *O1s*, and *N1s* spectra were collected with an analyzer pass energy of 40 eV. The binding energy (BE) scales were calibrated by setting the *C1s* BE to 285.0 eV. All spectra were fitted using the CasaXPS (version 2.3.17PR 1.1) software. Integration of the intensities of the XPS peaks was used to calculate element atomic percentages, considering the atomic sensitivity factors of the instrument data system.

#### 2.4.5. Fluorescence Spectroscopy

The amount of immobilized peptide was quantified through a colorimetric reaction using 9,10-phenanthrenequinone (PHQ, Fluka) following an adaptation of the work by Kazemzadeh-Narbat et al. [30]. This method is based on a reaction between PHQ and arginine present in Dhvar5 to obtain a stable fluorescent compound [31]. Briefly, samples were dissolved by sonicating the substrates for 1 h in 0.1 M aqueous HCl (1 mL) in an ultrasound bath. Afterward, 3 mL of 3.5 mM PHQ in absolute ethanol was added to 1 mL of each sample solution, followed by the addition of 0.5 mL of 2 M aqueous NaOH. The reaction was stopped after 3 h of incubation at 30 °C, using 2.25 mL of 2.4 M aqueous HCl. The fluorescence emission was measured using an excitation wavelength of 256 nm and detecting the emission at 380 nm in a fluorescence microplate reader (Biotek Synergy Mx Luminometer, BioTek Instruments, Winooski, VT, USA). Standard solutions of free *L*-arginine and free Dhvar5 were used to obtain calibration curves. The Dhvar5 calibration curve was adjusted by quantification of free peptide at 280 nm in a Thermo Scientific NanoDrop^®^ 1000 spectrophotometer (ThermoFischer Scientific, MA, USA). The amount of Dhvar5 in each film was then calculated based on those calibration curves.

#### 2.4.6. Atomic Force Microscopy (AFM)

AFM measurements were performed at CEMUP using a Multimode Nanoscope Iva microscope (Veeco, Plainview, NY, USA) with a 7236 EV scanner (ΔXY: 16 µm × 16 µm, ΔZ: 3.8 µm). A Bruker RTESP tip was used in tapping mode. Samples were imaged at three randomly chosen locations. The resulting images (2 µm × 2 µm) were treated and analyzed in the Nanoscope AFM Analysis software.

#### 2.4.7. Electrokinetic Analysis (EKA)

The zeta potential of both unmodified and modified chitosan films was determined using an ElectroKinetic Analyzer (Anton Paar GmbH, Graz, Austria) with a stamp cell suitable for flat substrates, based on the streaming potential method, as previously described by us [32,33]. Briefly, the stamp cell has two poly(methyl methacrylate) (PMMA) sample holders with a cross-section of 1 × 2 cm. A 1 mM KCl aqueous solution was used as the electrolyte and adjusted to pH 8.5, 6.6, 5.4, and 4.3. Analyses were carried out at room temperature, using four samples. During the assay, the electrolyte travels through a defined gap, adjusted by a micrometer screw, between the sample surfaces. This creates a differential pressure to disrupt the diffuse part of the electrochemical double layer established at the sample–electrolyte interface [34]. The streaming potential is then measured at the extremities of the streaming channel by Ag/AgCl electrodes. The streaming potential measurements were carried out while applying an electrolyte flow with alternating direction and pressure variation from 0 to 400 mbar. For each pH, a total of 12 measurements were performed (6 in each flow direction). The zeta potential was calculated by applying the Fairbrother–Mastin method.

#### 2.4.8. Quartz Crystal Microbalance with Dissipation (QCM-D)

Protein adsorption to Dhvar5-chitosan coatings was evaluated using a QCM-D system (Q-Sense E4 instrument; Biolin Scientific, Gothenburg, Sweden) and gold-coated QCM-D sensors with a fundamental frequency of 5 MHz (Biolin Scientific), as previously described [35]. Briefly, after cleaning, the sensors were prepared as described above (Section 2.2) [14]. Adsorption of bovine serum albumin (BSA) to Dhvar5-Chitosan surfaces was followed in real-time by the frequency and dissipation shifts of the sensors. A baseline was established by pre-incubating samples with phosphate-buffered saline (PBS, pH = 7.4) for 15 min at a flow rate of 0.1 mL/min. Then, a BSA solution at 4 mg/mL in PBS was injected into the system at a flow rate of 25 µL/min until saturation was achieved (2 h). PBS was then used to remove loosely attached proteins. The temperature was kept at 37 °C throughout the assay. The resulting data was treated with the Voigt model. Three replicates per sample were analyzed in three independent assays (n = 3).

### 2.5. Antibacterial Activity Assays

#### 2.5.1. Bacterial Strains, Media, and Growth Conditions

These assays were carried out using *Staphylococcus aureus* (*S. aureus*, ATCC 49230), *Staphylococcus epidermidis* (*S. epidermidis*, ATCC 35984), *Escherichia coli* (*E. coli*, ATCC 25922) and *Pseudomonas aeruginosa* (*P. aeruginosa*, ATCC 27853). Bacteria were firstly grown on Tryptic Soy Agar (TSA) and then overnight on Tryptic Soy Broth (TSB) at 37 °C, 150 rpm. Bacterial suspensions were adjusted by measuring Optical Density (600 nm). Bacterial numbers were confirmed by a retrospective viable count of colony-forming units (CFU).

#### 2.5.2. Surface Antimicrobial Activity Evaluation

All samples were kept in 70% (*v*/*v*) ethanol solution for 30 min, washed 3 times in sterile type 2 water (water with a resistivity greater than 1 MΩ-cm, a conductivity inferior to 1.0 µS/cm and less than 50 ppb of total organic carbon) and then dried in a sterile environment. Samples were then placed onto flat-bottom 24-well cell suspension culture plates (Sarstedt, Nümbrecht, Germany). Bacterial suspensions (5 μL, 10^8^ CFU/mL) in PBS were deposited onto the surface of each sample and then covered with a glass coverslip to ensure contact between bacteria and surface and incubated at 37 °C for 5 h. To avoid medium evaporation surrounding wells were filled with sterile type 2 water. Substrates were rinsed with 0.9% (*w*/*v*) sterile aqueous NaCl and then stained with a combination dye of the LIVE/DEAD^®^ Bacterial Viability Kit (Baclight™; Syto9/propidium iodide (PI)) for 15 min in the dark. Images in eight fields on each of the triplicate replicates were obtained with an inverted fluorescence microscope (Axiovert 200M, Zeiss, Jena, Germany) with a 400× magnification, corresponding to a net area of about 0.0946 mm^2^ per sample. Green cells were assumed to be alive, whereas red cells were considered dead, as the fluorescent emission of Syto9 is quenched by PI. The bacteria count was performed using the manual counting software included in the ImageJ software.

## 3. Results

### 3.1. Characterization of Dhvar5-Chitosan Coatings (Film and Bulk)

The overall procedure for the development of Dhvar-5 coatings (film) is described in Figure 1. After converting amines to azides in the chitosan films (Figure 1A), alkyne-modified Dhvar5 was immobilized onto the films in either orientation (Figure 1B1), resulting in the Dhvar5-chitosan coatings shown in Figure 1B2.

The success of Dhvar5 conjugation onto chitosan ultrathin coatings was confirmed by alterations in coating thickness (ellipsometry analysis), wettability (WCA measurements), structure (IRRAS), and composition (XPS). Dhvar5 surface density was calculated by fluorescence spectroscopy using the Arg-PHQ reaction that generates a stable fluorescent adduct. In addition, AFM, EKA, and QCM-D analyses were used to compare roughness, zeta potential, and protein adsorption on coatings where Dhvar5 was conjugated to chitosan before (bulk) or after (film) their production. For these comparisons, Dhvar5 was immobilized only by its *C*-terminus, i.e., using *C*_t_-Dhvar5-Chitosan coatings prepared by either the bulk or the film peptide conjugation method.

#### 3.1.1. Ellipsometry

Chitosan coatings thicknesses before and after surface modification are presented in Figure 2. Chitosan coating had a thickness of 20 ± 0.5 nm. After incubation with Dhvar5, a steep increase in film thickness supports the success of peptide conjugation. Moreover, both peptide immobilization orientations had the same thickness.

#### 3.1.2. Water Contact Angles (WCA) Analysis

WCA of chitosan coatings determined by sessile drop and captive bubble methods are presented in Figure 3. As observed in Figure 3A, after Dhvar5 binding through its *C*-terminus (exposing its hydrophobic region), the WCA increased (θ_w_ = 64°) when compared to unmodified chitosan (θ_w_ = 59°). The same trend was observed for the captive bubble method (Figure 3B) as Dhvar5-modified film exhibited a WCA of 45° and the unmodified chitosan presented a WCA of 35°. On the other hand, when the peptide was immobilized through its *N*-terminus, samples showed a more hydrophilic behavior, namely θ_w_ = 53° for sessile drop (Figure 3A) and θ_w_ = 15° for captive bubble (Figure 3B), compared to unmodified chitosan (θ_w_ = 59° for sessile drop and θ_w_ = 35° for captive bubble). These results are consistent with the immobilization of the peptide exposing its hydrophilic domain, rich in positively charged amino acids.

#### 3.1.3. Fourier Transform Infrared Reflection-Absorption Spectroscopy (IRRAS)

IRRAS spectra of chitosan, *N*_3_-chitosan, and Dhvar5-Chitosan (film) ultrathin coatings are shown in Figure 4. Characteristic absorption bands of chitosan were present in the spectrum of unmodified chitosan (Figure 4A), as previously described [24,25,32,36,37]. Nonetheless, at 1080 cm^−1^, the absorption band associated with the C–O–C stretching vibration in the glucopyranose ring in chitosan monomers appeared slightly shifted compared to transmission IR spectra obtained from the solid suspensions (pellets) of ground chitosan in KBr (1076 cm^−1^) [24]. These slight differences between chitosan IR spectra obtained through different acquisition modes have been previously reported [38]. After the conversion of chitosan amines into azides, the characteristic azide peak at 2120 cm^−1^, assigned to the asymmetric *N^−^=N^+^=N^−^* stretching mode, was observed as one of the most intense peaks in the spectrum of *N*_3_-chitosan (Figure 4B) [39,40]. IRRAS spectra of the ultrathin films obtained by the subsequent covalent conjugation of *C*_t_-Ahx-Dhvar5 and *N*_t_-Ahx-Dhvar5 onto *N*_3_-chitosan are displayed in Figure 4C,D, respectively. Both spectra were consistent with a successful “click” reaction, as two relevant changes could be noted when compared to *N*_3_-chitosan: (i) disappearance of the azide band at 2120 cm^−1^, suggesting complete conversion of azide groups after reaction with the alkyne-modified peptides; and (ii) increase in the intensity of the amide I, II and III bands (1650–1390 cm^−1^) [41], strongly corroborating peptide grafting onto the films [42]. Noteworthy, the characteristic band of chitosan at 1080 cm^−1^ (*C–O–C*) was not observable, which suggests that a high quantity of peptide was attached to the ultrathin films, thus masking the detection of the characteristic peaks from the underlying chitosan layer. These results support that the conditions employed in the CuAAC reactions were highly efficient.

#### 3.1.4. X-ray Photoelectron Spectroscopy (XPS)

Chitosan, *N*_3_-chitosan, and Dhvar5-chitosan ultrathin coatings were further analyzed by XPS. This is a useful tool to track surface modification for both organic (e.g., graphene [43,44]) and inorganic (e.g., gold [14] or titanium [13]) substrates, as it powerfully analyses a wide area, compared to other surface characterization techniques (i.e., Auger electron spectroscopy and secondary ion mass spectroscopy), and provides sensitive chemical composition within 10 nm for all elements, except *H* and *He* [45]. The XPS survey of unmodified chitosan and *N*_3_-chitosan ultrathin coatings showed the presence of residual gold, likely due to some heterogeneity in surface thickness (<10 nm), which leads to the detection of the gold substrate beneath the polymer. Relevantly, no residual copper was detected on the Dhvar5-chitosan coatings, indicating that the washes with EDTA after peptide tethering via CuAAC were sufficient to fully deplete the copper catalyst. Table 1 shows the relative atomic composition of samples regarding carbon, oxygen, and nitrogen.

As expected, the conversion of chitosan amines to azides (*N*_3_-chitosan) led to an increase in the percentage of nitrogen (*N1s*). In the Dhvar5-chitosan coatings subsequently prepared, the percentage of oxygen (*O1s*) decreased, while those of both carbon and nitrogen increased, which is consistent with successful tethering of the peptides via the triazole link formed upon CuAAC of the alkynyl-peptides with *N*_3_-chitosan. In the high-resolution XPS analysis of *C1s* (Figure 5), the spectrum of unmodified chitosan (Figure 5A) was resolved in three peaks, as previously described [36]. The peak at 285.0 eV was assigned to *C*–*C* and *C*–*H* type carbons that are typically related to adventitious carbon [36], the peak at 286.6 eV was assigned to *C–NH_*2*_*, *C–OH* and *C–O–C* carbons, and the peak at 288.3 eV was assigned to carbons from the *O–C–O* and *N–C=O* groups. The high-resolution XPS spectrum of *N1s* for the same ultrathin film (unmodified chitosan) showed a peak at 399.5 eV, assigned to nitrogen in *C–N* and *CO–N* bonds. No significant changes were observed in the high-resolution XPS spectrum of *C1s* for *N*_3_-chitosan (Figure 5B), whereas the high-resolution *N1s* XPS spectrum for this same coating exhibited, besides two peaks at 399.5 and 401.3 eV, respectively, for chitosan *NH*_2_ and *NH*_3_*^+^*, a peak at 404.3 eV associated with nitrogen from the azide group. In other words, the successful conversion of the chitosan amines into azides was further corroborated by these analyses [42,46]. The high-resolution XPS spectra of *C1s* and *N1s* for both Dhvar5-chitosan coatings (Figure 5C,D) were similar to each other, suggesting that immobilization occurred as expected, independently of peptide orientation. An increase in the 285.0 eV peak (*C–C* and *C–H*) was observed relative to the unmodified chitosan and *N*_3_-chitosan, suggesting the insertion of carbon atoms (*C–C* and *C–H*) present in the peptide chain. Additionally, the peak at 400.2 eV in the *N1s* spectrum, attributed to nitrogen on the triazole ring, agrees with the spectra in the literature [42,46]. Moreover, the peak assigned to nitrogen in the azide group (404.3 eV; observed in Figure 5B) disappeared, while the peaks from nitrogen in *C–N* and *CO–N* bonds and protonated amine groups remained present. Therefore, high-resolution spectra of *N1s* also demonstrate that all azides in *N*_3_-chitosan reacted with the alkyne-modified peptides via the CuAAC, establishing a triazole link between the peptide moiety and the chitosan layer. Furthermore, atomic percentages of the distinct types of nitrogen were analyzed, as summarized in Table 2. Conversion of amines to azides in chitosan is demonstrated by the appearance of a peak at 404.5 eV attributed to nitrogen in azide groups. The success of the subsequent CuAAC reaction was confirmed by the replacement of the previous band with a peak at 400.0 eV, which further supports the formation of the triazole ring. Altogether, these results corroborate the success of all reactional steps toward the synthesis of the desired peptide-chitosan conjugates.

#### 3.1.5. Peptide Surface Density Determination

Peptide surface densities were 60 ± 13 and 63 ± 16 µg/mm^2^ for the *C*_t_-Dhvar5-Chitosan and the *N*_t_-Dhvar5-Chitosan films, respectively. Thus, a similar average peptide density was obtained for both Dhvar5 derivatives, regardless of the orientation for peptide immobilization.

#### 3.1.6. Atomic Force Microscopy (AFM)

The surface roughness of Dhvar5-Chitosan coatings (film or bulk) was analyzed by AFM. Figure 6 shows that the surface roughness of the *C*_t_-Dhvar5-Chitosan coating (film) was six times higher (Ra ≈ 6.39 nm) than those of unmodified chitosan (Ra ≈ 1.00 nm) and *N*_3_-chitosan (Ra ≈ 0.85 nm). As expected, this was not observed for *C*_t_-Dhvar5-Chitosan (bulk) surfaces, where the roughness (Ra ≈ 0.85 nm) was similar to that of unmodified chitosan. All samples exhibited a homogeneous roughness profile across the imaged areas, as shown by the minimal standard deviation.

#### 3.1.7. Electrokinetic Analysis (EKA)

Surface charge was inferred from zeta potential determinations at four different pH values (8.5, 6.6, 5.4, and 4.3) (Figure 7). All zeta potential values increased with decreasing pH, as expected. As pH decreases, amines in chitosan and peptide side chains get protonated. However, chitosan and Dhvar5-chitosan ultrathin films still exhibit a negative zeta potential through most of the pH values tested due to the influence of the negative underlying gold layer. The bare gold (Au) surface, used as control, was always the most negatively charged surface over the entire pH range used. Zeta potential was significantly less negative for *C*_t_-Dhvar5-Chitosan (film) (ζ = −26 ± 2 mV) than for *C*_t_-Dhvar5-Chitosan (bulk) (ζ = −37 ± 2 mV) at pH 8.5. For pH 6.6 and 5.4, zeta potentials for chitosan and Ct-Dhvar5-Ch (film and bulk) were similar (ζ between −15 ± 5 and −19 ± 3 mV at pH 6.6; ζ between −6 ± 0 and −11 ± 2 mV at pH 5.5). At pH 4.3, *C*_t_-Dhvar5-Chitosan (film) exhibited a more negative zeta potential (ζ = −5 ± 1 mV) than both chitosan (ζ = 8 ± 1 mV) and *C*_t_-Dhvar5-Chitosan (bulk) (ζ = 10 ± 0 mV) coatings, which were similar to each other.

#### 3.1.8. Quartz Crystal Microbalance with Dissipation Monitoring (QCM-D)

The anti-fouling behavior of *C*_t_-Dhvar5-Chitosan ultrathin coatings prepared by peptide conjugation after (film) and the before (bulk) coating was evaluated by the quantification of protein adsorption using QCM-D. Figure 8 shows the adsorption of BSA per cm^2^ of the sample. The conjugation of *C*_t_-Dhvar5 on chitosan coatings *C*_t_-Dhvar5-Chitosan (film) induced more protein adsorption (≈579 ng/cm^2^) compared to chitosan (≈273 ng/cm^2^). On the opposite, films prepared using pre-synthesized Dhvar5-chitosan conjugates (*C*_t_-Dhvar5-Chitosan (bulk)) led to a significant reduction (>70%) in protein adsorption (≈77 ng/cm^2^), relatively to chitosan.

### 3.2. Antibacterial Activity Assays

#### Surface Antimicrobial Activity Mechanism

Evaluation of bacterial viability using LIVE/DEAD^®^ Bacterial Viability Kit (Baclight™, Invitrogen, Waltham, MA, USA) is presented in Figure 9. As expected, a sharp reduction in total adhered bacteria (~40%) between unmodified chitosan and bare Au was observed, in agreement with previously reported antimicrobial properties of chitosan ultrathin films [14]. Regarding antimicrobial activity against *S. aureus* (Figure 10), samples with covalently immobilized peptide exhibited a pronounced decrease of bacterial adhesion as compared to chitosan, reaching a ~60% reduction when the peptide was immobilized through its *C*-terminus (*C*_t_-Dhvar5-Chitosan (film)) (*p* < 0.05). Only a slightly lower (~52%) reduction of bacterial adhesion was achieved when the peptide was tethered through its *N*-terminus (*N*_t_-Dhvar5-Chitosan (film)) (*p* < 0.05). A similar antiadhesive effect was observed against the other bacteria tested, although it was less pronounced in Gram-negative bacteria. It was also observed that, for all surfaces analyzed, most of the adhered bacteria were not dead. Still, surfaces with *C*-terminally immobilized Dhvar5 were the most antiadhesive, with a two-fold lower number of live adhered bacteria than those in unmodified chitosan.

## 4. Discussion

In this work, Dhvar5 was covalently immobilized onto chitosan ultrathin films using the CuAAC “click” chemistry to evaluate if this strategy could be effective in creating coatings able to prevent bacterial colonization. Dhvar5 was grafted in different orientations since the importance of peptide orientation on its antimicrobial behavior was reported by us for this peptide [1,14] and by others for other peptides [20,47,48]. The successful chemoselective conjugation of Dhvar5 onto chitosan coatings in a controlled orientation was confirmed by a combination of surface characterization techniques. WCA measurements showed the expected increase in surface hydrophobicity when Dhvar5 was conjugated through its *C*-terminus (i.e., exposing its hydrophobic region). Conversely, hydrophilicity increased when Dhvar5 was conjugated through its *N*-terminus, exposing its cationic region. Therefore, the chemoselective orientation-controlled grafting of the peptide onto the ultrathin coatings was clearly demonstrated. Moreover, the comparable increase of the coating thickness after Dhvar5 conjugation in both orientations (determined by ellipsometry) suggested a similar Dhvar5 surface density independent of peptide orientation. This fact was confirmed by fluorometric quantification of grafted peptide, which was similar for both *C_t_*-Dhvar5-Chitosan and *Nt*-Dhvar5-Chitosan films. These observations were fully corroborated by IRRAS and XPS analyses. Hence, the increased intensity of the characteristic peptide amide I band at 1650 cm^−1^ in the IRRAS spectra of the peptide-grafted films, alongside the disappearance of the azide band 2120 cm^−1^ that was present in the *N*_3_-chitosan coating, demonstrate the efficient peptide tethering onto this coating via CuAAC. This was further supported by the appearance of the nitrogen characteristic peak of the triazole ring (400.2 eV) in the XPS spectra of the peptide-tethered coatings, in replacement of the azide group (404.3 eV) observed in the spectrum of the precursor *N*_3_-chitosan coating [14,24].

Regarding antimicrobial activity, both the *C*_t_-Dhvar5-Chitosan and the *N*t-Dhvar5-Chitosan (film) coatings herein reported showed a significant reduction in bacterial adhesion compared to unmodified chitosan, highlighting the advantage of AMP covalent conjugation. This antiadhesive effect was higher in *C*_t_-Dhvar5-Chitosan, except for *S. aureus*, where no influence of peptide orientation could be detected. Interestingly, a higher antiadhesive effect was previously described by us for *S. aureus* when Dhvar5 was grafted onto chitosan coatings through its *N*-terminus [1]. However, in this previous study, a distinct bacterial strain was used (methicillin-resistant *S. aureus*; ATCC 33591), and also a different type of immobilization chemistry (formation of a disulfide bridge), which resulted in a much lower peptide density (>25,000 times lower) than that herein achieved via CuAAC [1]. In line with this, we have recently tested the grafting of Dhvar5 onto chitosan through the highly efficient thiol-norbornene photo-click chemistry (TNPC), and ultrathin films thus obtained showed a 35% reduction of total adhered Gram-positive *S. epidermidis* [19]. Interestingly, coatings previously prepared by us using pre-synthesized Dhvar5-chitosan conjugates (bulk) displayed bactericidal effects against Gram-positive bacteria (*S. aureus* and *S. epidermidis*) [14].

Concerning Gram-negative bacteria (*E. coli* and *P. aeruginosa*), the Dhvar5-grafted coatings herein reported also show a higher antiadhesive effect than chitosan, in line with our previous observations with coatings prepared using pre-synthesized Dhvar5-chitosan conjugates (bulk) [14]. Relevantly, the peptide-tethered ultrathin films we recently prepared via TNPC showed an increased adhesion and killing of Gram-negative *P. aeruginosa* compared to chitosan [19].

It has been previously discussed that peptides with up to two arginines, such as Dhvar5, have a lower affinity towards the membranes of Gram-negative bacteria than towards those of Gram-positive ones [14,49]. Moreover, the net charge of Dhvar5 and the hydrophobicity of some of its residues, namely leucine and phenylalanine, may also contribute to the peptide’s higher activity towards Gram-positive as compared to Gram-negative bacteria [50]. In order to better understand the differences in the antimicrobial effect of Dhvar5-grafted chitosan films towards Gram-positive bacteria, film and bulk coatings were produced and compared using several characterization techniques. For these assays, Dhvar5 was only grafted through its *C*-terminus.

No differences in surface charge (zeta potential) were detected between the coatings at physiological pH at which the bacterial assays were carried out. Therefore, differences in antimicrobial action cannot be explained by distinct surface charges.

The higher nano-roughness obtained for the *C*_t_-Dhvar5-Chitosan (film) coating as compared to *C*_t_-Dhvar5-Chitosan (bulk) showed that, as expected, peptide molecules were more exposed on the former than on the latter. Moreover, when Dhvar5 was grafted on a pre-cast chitosan coating (film), peptide density was almost 35 times higher than that achieved for films prepared using pre-synthesized Dhvar5-chitosan conjugate (bulk) [14]. Previous reports highlighted the influence of peptide surface density on its antimicrobial activity. For instance, Chen et al. [47] found that higher peptide concentration was associated with stronger antibacterial activity. Humblot et al. [51] related low peptide concentrations with bacteriostatic rather than bactericidal effect, probably because low peptide densities do not allow multiple neighboring peptides to contact the cell membrane. Moreover, Hilpert and co-workers [48] described that antimicrobial activity decreased with the concentration of the immobilized peptide. Interestingly, our work using *C*_t_-Dhvar5-chitosan coatings (film vs. bulk) shows that the bactericidal effect on Gram-positive bacteria previously reported for the lower-density (bulk) coating [14] is replaced by an antiadhesive behavior when peptide density is very high (film). Nonetheless, immobilization chemistries are also relevant for this bactericidal vs. antiadhesive action, as we also had previously observed antiadhesive effects for Dhvar5-Chitosan thin films with much lower peptide density (bis-thiol/disulfide immobilization chemistry) [1] or similar peptide concentration (TNPC) [19] as in CuAAC bulk. Thus, differences in antimicrobial effect cannot be explained solely by different peptide surface densities. Interestingly, the antiadhesive effect observed for bacteria was not observed for protein adsorption since *C*_t_-Dhvar5-chitosan coatings (film) adsorbed twice as much BSA compared to chitosan. The expected anti-fouling behavior was only observed on *C*_t_-Dhvar5-conjugates (bulk) coatings, with a 70% reduction of BSA adsorption compared to chitosan. These differences must be related to Dhvar5 density and exposure on the surface. When Dhvar5 was immobilized on pre-cast chitosan coatings (film), the peptide was grafted onto the outermost layers of the coating at high density and exposing its hydrophobic terminus, which increased both the nano-roughness and hydrophobicity of the surface, therefore favoring protein adsorption [52]. The high nano-roughness could explain the antiadhesive behavior of bacteria without impairing protein adsorption, although both protein and bacterial adhesion are often enhanced by an increase in roughness [53,54]. However, it has been highlighted that micro- to nano-roughness can be uninviting for bacterial adhesion, regardless of surface hydrophobicity/hydrophilicity [55,56]. In contrast, when coatings were prepared using *C*_t_-Dhvar5-conjugates (bulk), their surface physicochemical properties (wettability [14], nano-roughness, and charge at physiological pH) were similar to those of chitosan. The low protein adsorption on these coatings (bulk) may be explained by the partial occlusion of peptide molecules within the chitosan layer, somehow favoring both protein anti-fouling and bacterial killing effects. Although anti-fouling to proteins, *C*_t_-Dhvar5-chitosan (bulk) surfaces did not avoid bacterial adhesion; rather, they were able to bind and kill Gram-positive bacteria (>50% adherent *S. aureus* and >75% adherent *S. epidermidis*) [14].

Altogether, our previous [1,16,36] and current findings clearly highlight the multifactorial nature of the antimicrobial behavior displayed by peptide-tethered biopolymers. Our results with the Dhvar5-chitosan coatings herein show that such behavior depends at least on the bacteria (species and strain), peptide-to-surface immobilization chemistry, peptide surface density and orientation, as well as surface roughness.

## 5. Conclusions

This work demonstrates that the covalent immobilization of Dhvar5 onto a chitosan ultrathin coating, using the chemoselective CuAAC “click” reaction, produces a peptide-tethered surface showing decreased colonization by both Gram-positive and Gram-negative bacteria. This antiadhesive effect was higher when Dhvar5 was grafted through its *C-terminus*, exposing its hydrophobic region. A comprehensive comparative study of the surface properties of the coatings herein developed with those previously reported by us, where the ultrathin coating was produced using pre-synthesized Dhvar5-chitosan, shows that multiple parameters influence the antibacterial potency and mode of action of the surfaces. Relevantly, bacterial antiadhesive effects may be at least partly explained by the high nano-roughness of the high-density peptide surfaces herein reported. In conclusion, by conveniently harnessing relevant parameters that modulate antimicrobial action, Dhvar5-chitosan materials can be advanced, which holds great promise for developing effective antimicrobial coatings for application in the biomedical field.

## Figures and Tables

**Figure 1 pharmaceutics-15-01510-f001:**
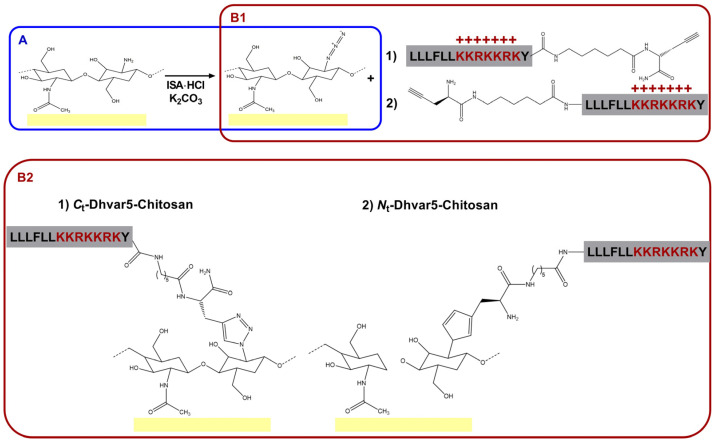
(**A**) Chitosan functionalization by direct conversion of the amines in the chitosan ultrathin film coatings into azides (*N*_3_-chitosan); (**B1**) CuAAC “click” reaction between the azide group on *N*_3_-chitosan and the alkyne group in the Pra residue inserted at either the peptide’s (**1**) C-terminus or (**2**) N-terminus, to originate (**B2**) the Dhvar5-grafted chitosan films (**1**) *C*_t_-Dhvar5-Chitosan and (**2**) *N*_t_-Dhvar5-Chitosan.

**Figure 2 pharmaceutics-15-01510-f002:**
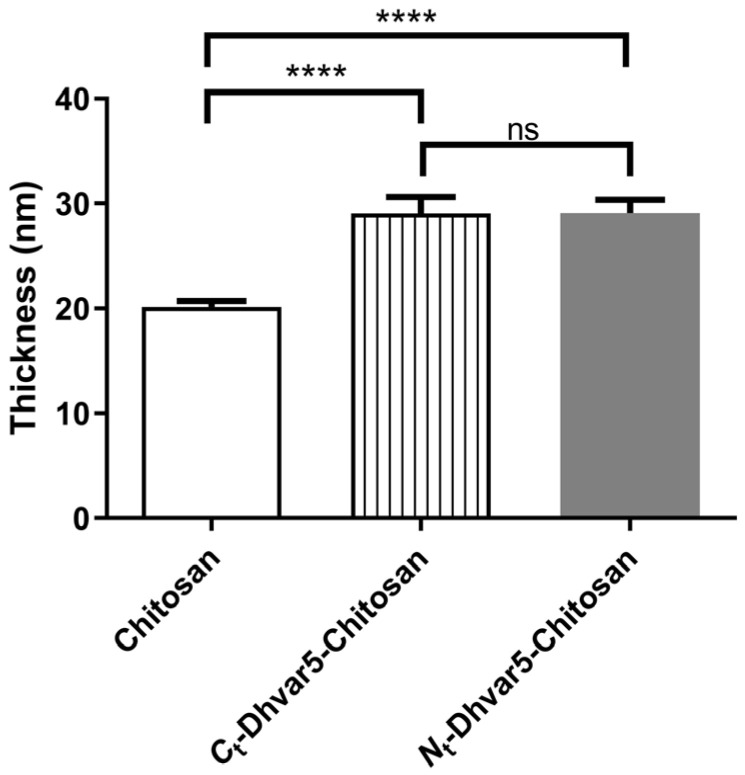
Ellipsometry analysis of the chitosan and chitosan-functionalized films (One-Way ANOVA **** *p*  <  0.0001); data represent mean  ±  standard deviation; ns: non significant.

**Figure 3 pharmaceutics-15-01510-f003:**
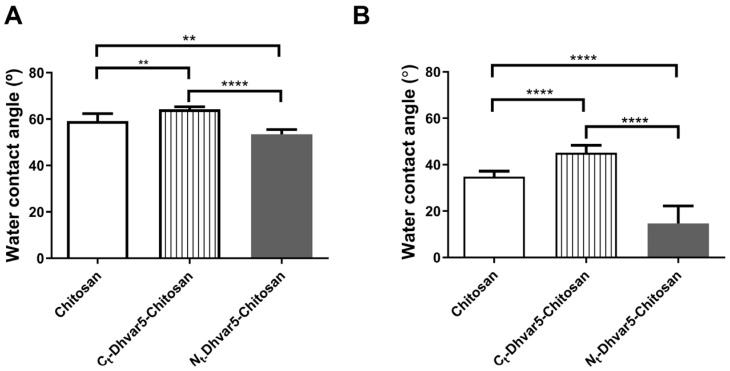
WCA for chitosan and chitosan-functionalized films determined by (**A**) sessile drop and (**B**) captive bubble methods (One-Way ANOVA analysis, ** *p* < 0.004, **** *p* < 0.0001); data represent mean  ±  standard deviation.

**Figure 4 pharmaceutics-15-01510-f004:**
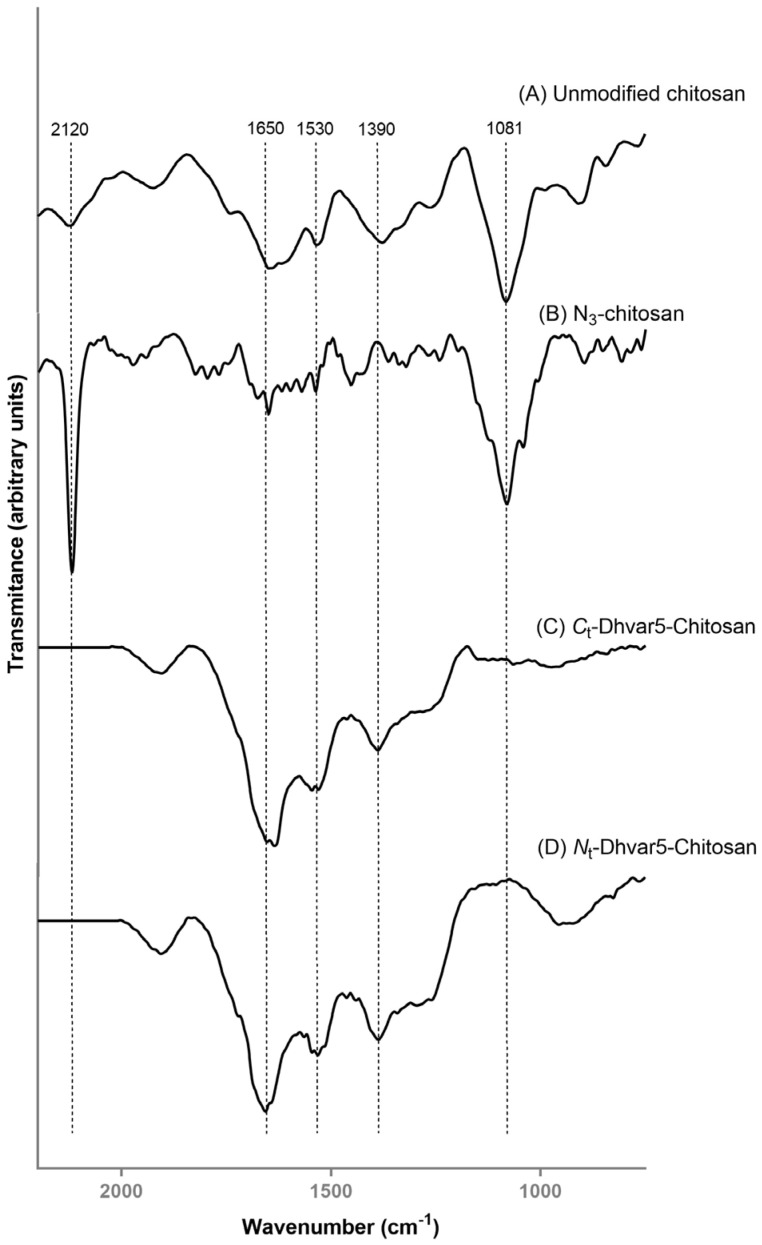
IRRAS spectra of (A) unmodified chitosan, (B) *N*_3_-chitosan, (C) *C*_t_-Dhvar5-Chitosan, and (D) *N*_t_-Ahx-Dhvar5-Chitosan ultrathin coatings. Dashed lines at 1650, 1530, and 1390 cm^−1^ correspond to amide I (C=O stretching), amide II (N–H bending), and amide III (C–N stretching vibrations), respectively. The dashed line at 1081 cm^−1^ is associated with stretching vibration C–O–C in the glucopyranose ring. The relative intensity of these (amide versus C–O–C bands) undergoes the expected evolution as a consequence of the entry of the peptide chains. The dashed line at 2120 cm^−1^ denotes a change in the azide band as a consequence of film modifications.

**Figure 5 pharmaceutics-15-01510-f005:**
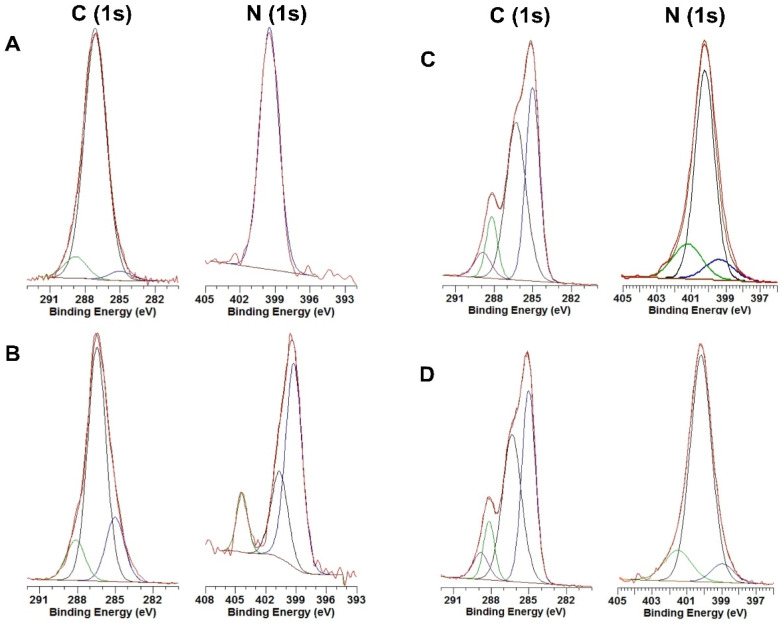
XPS high-resolution spectra of (**A**) unmodified chitosan, (**B**) *N*_3_-chitosan, (**C**) *C*_t_-Dhvar5-Chitosan, and (**D**) *N*_t_-Dhvar5-Chitosan, for *C1s* and *N1s* regions.

**Figure 6 pharmaceutics-15-01510-f006:**
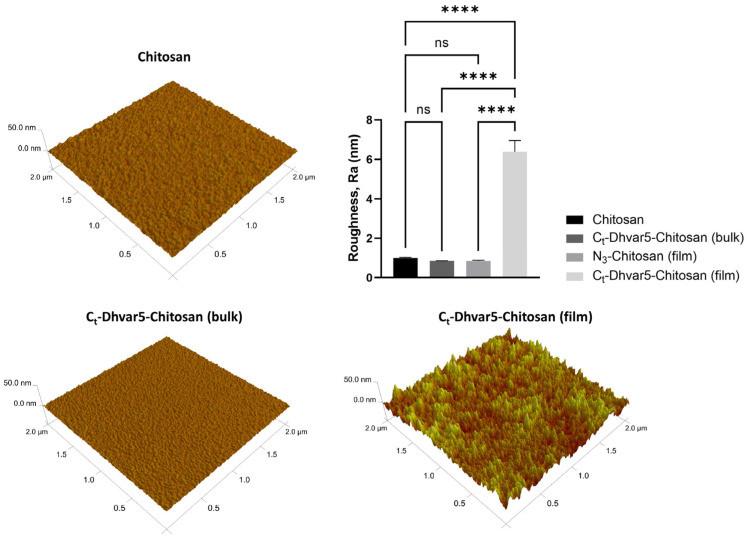
Atomic force microscopy (AFM) micrographs to evaluate surface topography and respective roughness (Ra) quantification (One-way ANOVA, **** *p*  <  0.0001); data represent mean  ±  standard deviation; ns: non-significant.

**Figure 7 pharmaceutics-15-01510-f007:**
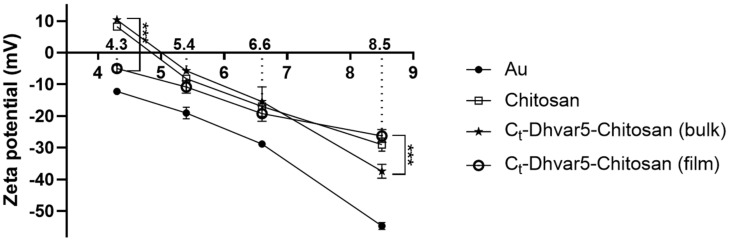
Zeta potential determination by EKA at varying pH values (Kruskal–Wallis test, *** *p* < 0.001); data represent mean  ±  standard deviation.

**Figure 8 pharmaceutics-15-01510-f008:**
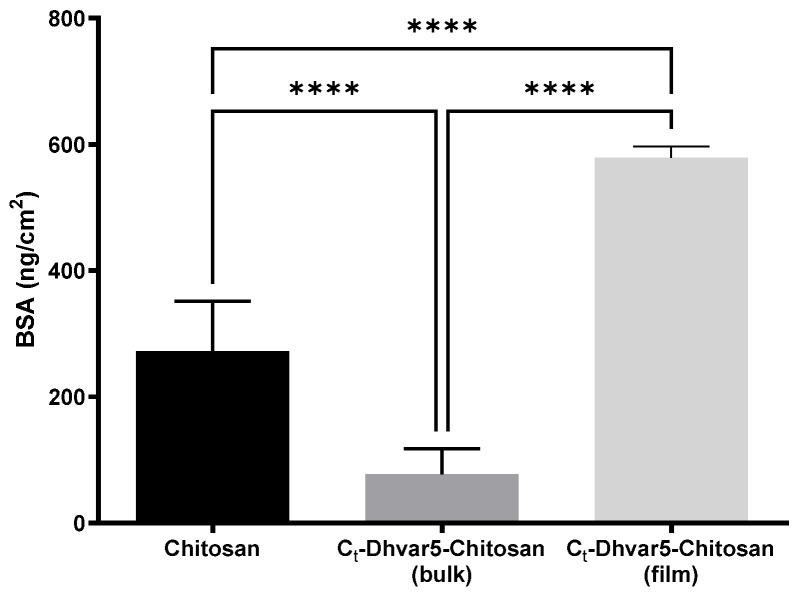
Adsorption of BSA (4 mg/mL solution in PBS, injected in the system at a flow rate of 25 µL/min) onto unmodified and peptide-modified chitosan films before and after film fabrication (Kruskal–Wallis test, **** *p*  <  0.0001); data represent mean  ±  standard deviation.

**Figure 9 pharmaceutics-15-01510-f009:**
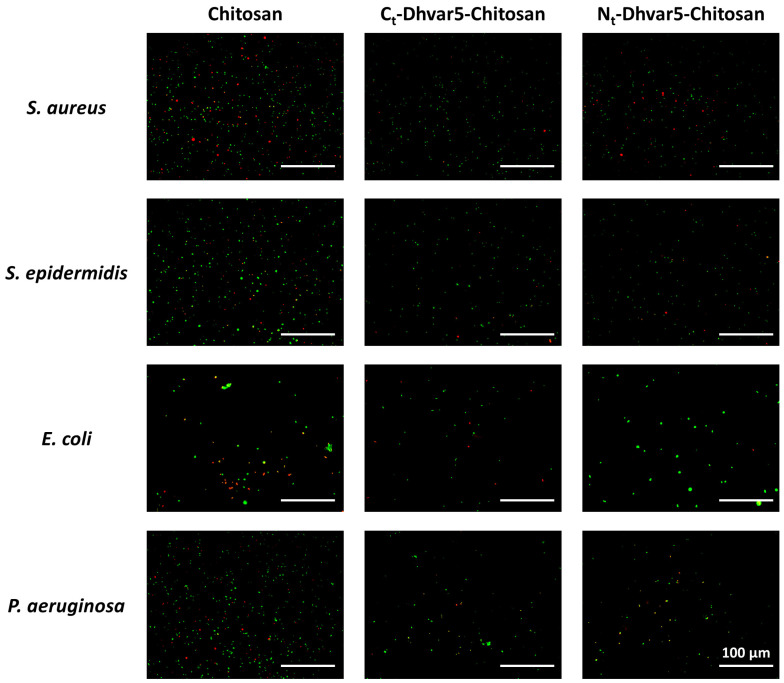
Representative images of the LIVE/DEAD^®^ Bacterial Viability Kit (Baclight^TM^) staining of adhered bacteria on the surface of the prepared samples. An inverted fluorescence microscope was used with a magnification of 400×. The scale bar corresponds to 100 µm.

**Figure 10 pharmaceutics-15-01510-f010:**
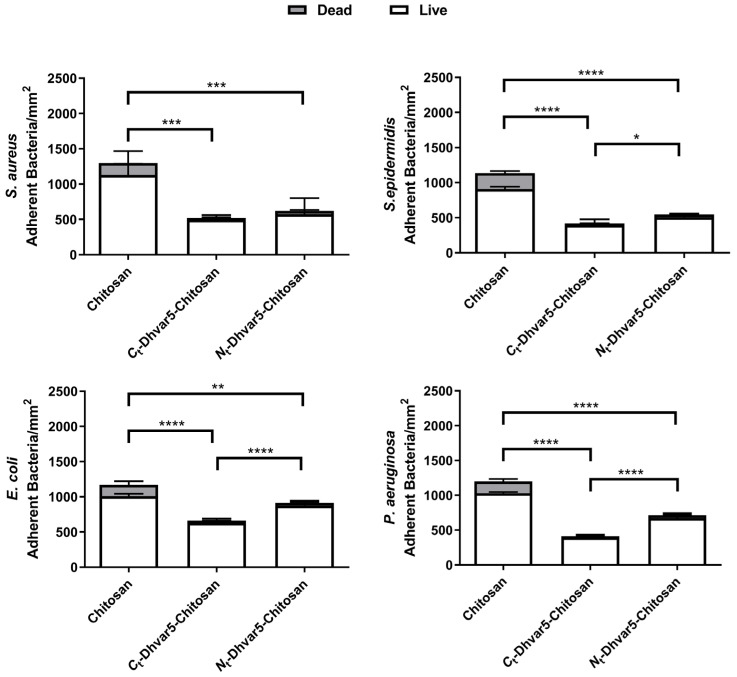
Viability of adhered *S. aureus*, *S. epidermidis*, *E. coli*, and *P. aeruginosa* incubated at 37 °C for 5 h (Two-way ANOVA * *p* < 0.02, ** *p* < 0.009, *** *p* < 0.0008, **** *p*  <  0.0001); data represent mean  ±  standard deviation. The gray area represents the number of dead bacteria.

**Table 1 pharmaceutics-15-01510-t001:** Elemental analysis data (% *C*, *N*, *O*) as determined by XPS analysis of unmodified chitosan thin film and respective derivatives.

Polymer	Atomic Composition (%)		
	*C1s*	*N1s*	*O1s*
Chitosan	56.9	8.3	34.8
*N*_3_-chitosan	56.5	10.4	33.1
*C*_t_-Dhvar5-Chitosan (film)	64.7	16.4	18.9
*N*_t_-Dhvar5-Chitosan (film)	65.3	16.4	18.3

**Table 2 pharmaceutics-15-01510-t002:** Chemical surface high-resolution analysis of *N1s* region for chitosan and respective derivatives.

Polymer	Atomic% *N1s*			
	*C* *–* *N/CO* *–* *N*	*CO* *–* *N* *–* *CO/N* *–* *CO* *–* *O*	*NH_3_^+^*	*N=N^+^=N^−^*
	399.5 eV	400.2 eV	401.3 eV	404.3 eV
Chitosan	100	-	-	-
*N*_3_-chitosan	61.7	-	26.2	12.1
*C*_t_-Dhvar5-Chitosan (film)	32.9	50.1	17.0	-
*N*_t_-Dhvar5-Chitosan (film)	6.70	79.8	13.5	-

## Data Availability

The raw data required to reproduce these findings are available upon reasonable request to the authors. The processed data required to reproduce these findings are available upon reasonable request to the authors.

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
