# Peer review of "Influence of Immobilization Strategies on the Antibacterial Properties of Antimicrobial Peptide-Chitosan Coatings"

_pharmaceutics, 2023, doi:10.3390/pharmaceutics15051510_

Round 1
Reviewer 1 Report
The authors report the functionalization of chitosan thin layers with the antimicrobial peptide Dhvar5 by click chemistry. The success of functionalization was confirmed by ellipsometry analysis, contact angle analysis, infrared spectroscopy and XPS.
The antimicrobial activity and surface properties of the produced coatings were compared to those of coatings fabricated using peptide-chitosan conjugates previously immobilized in bulk.
While the topic addressed in the manuscript is compelling, it is recommended that some sections be revised prior to publication.
1) I suggest to specify in the title that Dhvar5 is an antimicrobial peptide. Similarly, expand on the meaning of the acronym CuAA (Copper-catalysed azide-alkyne cycloadditions) in the abstract.
2) Please, widen the introduction to provide a brief overview of the key approaches for developing antimicrobial/antibiofilm surfaces. Some examples could include utilizing cationic compounds (https://doi.org/10.1021/acs.langmuir.2c02787), phenolic derivatives (https://doi.org/10.1039/D2PY01183B) and quorum sensing inhibitors (https://doi.org/10.2174/1381612820666140905114627).
3) Section 2.2.1: Which was the medium used for chitosan solubilization? Simply water?
4) On page 9, rather than attempting to deduce quantitative conversions from IR spectra, I recommend simply stating that the IR data supports the efficiency of the CuAAC reaction conditions.
5) Section 3.1.4: The versatility of XPS for tracking surface modification has been demonstrated in the literature for both organic and inorganic substrates. This fact could be highlighted by adding proper references (https://doi.org/10.3390/nano12010043 and https://doi.org/10.1016/j.apsadv.2022.100332).
6) Section 3.1.5: An explanation of the negative zeta potential of bare chitosan and AMP functionalized chitosan samples should be provided.
7) Antimicrobial activity: was there any indication of stain adsorption by chitosan when the author used a Live/dead kit to stain bacteria adherent to the chitosan? Are there any fluorescent images to show live and dead bacteria on chitosan coatings?
Reviewer 2 Report
The authors reported antimicrobial peptides immobilized chitosan coatings with good antimicrobial performance of the surface on Gram-positive bacteria and Gram-negative. The antiadhesive effect was not due to changes in surface wettability or protein adsorption, but it rather depended on variations in peptide concentration, exposure and surface roughness. The submission can be accepted after revision taking into account the following points:-
1. The photographs of Dhvar5-Chitosan coatings (film or bulk) should be provided.
2. Why the surface roughness of Ct-Dhvar5-Chitosan coating (film) was higher than Ct-Dhvar5-Chitosan coating (bulk)?
3. The fluorescence images of S. aureus, S. epidermidis, E. coli and P. aeruginosa incubated with the scaffolds and stained with LIVE/DEAD® Bacterial Viability Kit should be provided.
4. Normally, chitosan based materials hold inherent antimicrobial properties; the author should give more discussion on why further antimicrobial peptides immobilization is essential.
5. A comparison with previously published antimicrobial biomaterials should be discussed in introduction part. References for antimicrobial and chitosan scaffolds should be updated including these examples; Adv. Funct. Mater. 2023, 33, 2213342; Front. Bioeng. Biotechnol. 9:820434. doi: 10.3389/fbioe.2021.820434; RSC Adv., 2020, 10, 17280–17287.
6. The language should be revised and typos should be corrected.
The language should be revised and typos should be corrected.
Round 2
Reviewer 2 Report
Accept in present form